# Reproduction study: Towards Transparent and Explainable Attention Models

## 1 Reproducibility Summary

### 2 Scope of Reproducibility

Mohankumar et al. (2020) claim that current attention mechanisms in LSTM based encoders can neither provide a faithful nor a plausible explanation of the model's predictions in Natural Language Processing tasks. To make attention mechanisms more faithful and plausible, the authors propose two modified LSTM models with a diversity-driven training objective that ensures that the hidden representations learned at different time steps are diverse: the Orthogonal LSTM and the Diversity LSTM. The authors claim that the resulting attention distributions from these diversity-driven LSTMs offer more explainability and transparency in contrast to a Vanilla LSTM.

### 9 Methodology

The original code of the authors has been used. Data was retrieved from the links provided by the authors. A subset of the datasets used in the original paper were used, while maintaining the variety of different NLP tasks of the paper. The experiments were ran on the UvA Lisa cluster computer. Depending on the dataset, training and evaluation took between 1 and 40 hours. Additionally, the LIME framework was added to the pipeline to account for an extra experiment.

### 14 Results

Our results partially support the authors' claims. Although we were not able to reproduce everything the authors claimed in their paper, there are still some signs that the proposed diversity-driven LSTMs could offer something extra in terms of explainability and transparency.

### 18 What was easy

The authors' code was relatively easy to run. Their clear instructions for setting everything up and running experiments contributed to this. Some slight adaptations to the code had to be made to omit warnings, but this was a straightforward task. Their choice to automatically run and plot all experiments after each other was convenient for reproducing the work.

### 23 What was difficult

Setting up the environment in the remote GPU was slightly difficult. Also, the links for some datasets were malfunctioning or missing, making it impossible to verify all results. Running some experiments took quite a long time, but this was no major issue given the available computational resources we had on the Lisa GPU server.

### 27 Communication with original authors

There has been no contact with the orignal authors of the paper.

# 1 Introduction

Attention mechanisms in neural network-based models play an important role in various Natural Language Processing (NLP) tasks. Studies on the interpretability of these attention distributions, have often led to the notion of faithful and plausible explanations for model predictions. A distribution can be considered faithful if a higher attention weight implies a greater impact on the model's prediction. A prediction can be considered plausible if it provides a human-understandable justification for the model's prediction. Mohankumar et al. (2020) state that current LSTM attention mechanisms provide neither faithful nor plausible explanations of the model's predictions. This is mainly due to hidden representations within the LSTM being very similar at different timesteps. According to Mohankumar et al. (2020), by modifying the LSTM cell with a diversity-driven training objective that ensures highly dissimilar hidden representations, attention mechanisms can be made more faithful and plausible. The authors propose two different LSTM models to ensure high diversity between hidden states: a Diversity LSTM and an Orthogonal LSTM.

# 2 Scope of reproducibility

## 2.1 Target claims

In the paper, the authors introduce modified LSTM cells which makes attention distributions more faithful and plausible on NLP tasks including Binary Classification, Natural Language Inference, Paraphrase Detection and Question Answering. The main claims of the authors to substantiate this statement are as follows:

1. The predictive performance of the Diversity and Orthogonal LSTM models is comparable to that of a Vanilla LSTM.
2. The resulting attention distributions of the diversity-driven LSTM models offer more transparency as they provide a more precise importance ranking of the hidden states.
3. The resulting attention distributions of the diversity-driven LSTM models offer more transparency as they are better indicative of words important for the model's predictions.
4. The resulting attention distributions of the diversity-driven LSTM models offer more transparency as they correlate better with attribution methods such as gradients and integrated gradients.

By reproducing a series of experiments, the aim of this study is to determine which claims are supported. Due to the stochastic nature of the experiments that will be run to support or oppose these claims, identical numerical results are not expected. Instead, our judgement will be applied to determine which claims are supported by our results.

# 3 Methodology

## 3.1 Model descriptions

In this section we will briefly outline the theories behind the proposed Orthogonal and Diversity LSTM models. The information provided in this section stems from the original Mohankumar et al. paper. First the concept of conicity will be explained. Thereafter, the theory behind both LSTM models is made clear.

### 3.1.1 Conicity

One of the similarities measures defined in the original paper is conicity. This measures the similarity between a set of vectors. To do this, the alignment to a mean is calculated for a vector $v_i$ as follows:

$$\text{ATM}\left(\mathbf{v}_i, \mathbf{V}\right) = \text{cosine}\left(\mathbf{v}_i, \frac{1}{m} \sum_{j=1}^{m} \mathbf{v}_j\right).$$

The conicity is then defined as the mean of ATM of all vectors.

$$\text{conicity}\left(\mathbf{V}\right) = \frac{1}{m} \sum_{i=1}^{m} \text{ATM}\left(\mathbf{v}_i, \mathbf{V}\right)$$

A high conicity indicates high similarity between vectors and thus that they lie within a narrow cone centered at the origin. We will use the conicity measurement to evaluate the similarity between hidden states in the different LSTM models.

### 3.1.2 Orthogonal LSTM

With the Orthogonal LSTM, low conicity between hidden states of the LSTM encoder is ensured by orthogonalizing the current hidden states with the mean of previous states.

The Orthogonal LSTM uses the same set of update equations as a Vanilla LSTM where only the equation for updating the hidden states is interchanged with the following two equations:

$$\overline{\mathbf{h}}_t = \sum_{i=1}^{t-1} \mathbf{h}_i \qquad\qquad \mathbf{h_t} = \hat{\mathbf{h}}_t - \frac{\hat{\mathbf{h}}_t^T \overline{\mathbf{h}}_t}{\overline{\mathbf{h}}_t^T \overline{\mathbf{h}}_t} \overline{\mathbf{h}}_t$$

Where the hidden state vector's $\hat{\mathbf{h}}_t$ component is subtracted along the mean $\overline{\mathbf{h}}_t$ of the previous states.

### 3.1.3 Diversity LSTM

While the Orthogonal LSTM model sets a hard constraint between the hidden state vector's $\hat{\mathbf{h}}_t$ and the previous states mean $\overline{\mathbf{h}}_t$, the Diversity LSTM takes a more flexible approach by being trained to maximize the log-likelihood of the training data and minimize the conicity of hidden states.

$$L(\theta) = -p_{\text{model}}\left(y \mid \mathbf{P}, \mathbf{Q}, \theta\right) + \lambda \operatorname{conicity}\left(\mathbf{H}^P\right)$$

where $y$ is the ground truth class, $\mathbf{P}$ and $\mathbf{Q}$ are the input sentences, $\mathbf{H}^P = \{\mathbf{h}_1^p, \ldots, \mathbf{h}_m^p\} \in \mathbb{R}^{m \times d}$ contains all the hidden states of the LSTM, $\theta$ is a collection of the model parameters and $p_{\text{model}}(.)$ represents the model's output probability. $\lambda$ is a hyperparameter that controls the weight given to diversity in hidden states during training.

### 3.2 Datasets

In this study, experiments were reproduced for a subset of the original datasets. These datasets are the SST, IMDB, 20News, Yelp, SNLI, QQP and BabI datasets. The same preprocessing and splitting into subsets was performed as in the original paper. The code to download, preprocess and split the data can be found in the git of original paper: https://github.com/akashkm99/Interpretable-Attention. A full overview of all datasets can be found in Appendix A.

### 3.3 Hyperparameters

We have used the same hyperparameter-settings as in the Mohankumar et al. paper to be able to reproduce their results. For all datasets, except the BabI datasets, pretrained GloVe Pennington et al. (2014) or fastText Tomas et al. (2018) word embeddings have been used. The 50 dimensional BabI word-embeddings are learned form scratch during training. An overview of the hyperparameters can be found in appendix A.

### 3.4 Experimental setup

The link to our code has been made publicly available at https://github.com/Jeroenvanwely/FACT.git. We also provide the link to the code used in the Mohankumar et al. paper: https://github.com/akashkm99/Interpretable-Attention. We ran the code on the Lisa GPU server provided by the University of Amsterdam (see https://userinfo.surfsara.nl/systems/lisa for more information).

### 3.5 Computational requirements

Depending on the dataset, running experiments takes between 1 and 40 hours on a single node GPU. Therefore, a substantial amount of time is needed to complete experiments for all datasets. All code including datasets amounts to 15 GB, therefore this is the absolute minimum amount of storage that needs to be available.

## 4 Results

The results of our experiments support claims 1 and 2 from section 2.1. We did, however, not find strong evidence which supports claims 3 and 4. In the following sections we present the results of our experiments in detail. For each experiment, we will indicate if the results support or oppose a claim.

## 4.1 Experiment 1: Empirical evaluation

The first experiment performed in the original paper is an empirical evaluation (i.e. measurement of accuracy and conicity) of the Vanilla, Diversity and Orthogonal LSTM on different datasets. The results of the authors suggest that the accuracy of the Diversity and Orthogonal LSTM is similar to that of the Vanilla LSTM while conicity values decrease significantly.

Our obtained results regarding the empirical evaluation of the Vanilla, Diversity and Orthogonal LSTM averaged over three separate runs including the standard deviation are shown in table 1. For reference, the authors' results have been included as well. Compared to the authors' results, we see similar trends: the accuracy of the Diversity and Orthogonal LSTM is comparable to that of the Vanilla LSTM while the conicity decreases substantially. These findings support claim 1, which states that the predictive performance of the Diversity and Orthogonal LSTM models is comparable to that of a Vanilla LSTM.

| | LSTM | | Diversity LSTM | | Orthogonal LSTM | | Random | MLP |
|---|---|---|---|---|---|---|---|---|
| | Accuracy | Conicity | Accuracy | Conicity | Accuracy | Conicity | Conicity | Accuracy |
| **Binary Classification** | | | | | | | | |
| SST | **81.79** / **80.13** (1.06) | 0.68 / 0,72 (0.02) | 79.95 / 79.75 (0.86) | 0.20 / 0.19 (0.00) | 80.05 / 79.01 (0.62) | 0.28 / 0.28 (0.00) | 0.25 | 80.05 |
| IMDB | **89.49** / **89.86** (0.28) | 0.69 / 0.61 (0.04) | 88.54 / 87.23 (0.43) | 0.08 / 0.09 (0.00) | 88.71 / 88.44 (0.07) | 0.18 / 0.17 (0.01) | 0.08 | 88.29 |
| 20News | **93.55** / 90.29 (1.43) | 0.77 / 0.83 (0.06) | 91.03 / 92.16 (0.56) | 0.15 / 0.12 (0.00) | 92.15 / **92.31** (1.50) | 0.23 / 0.24 (0.01) | 0.13 | 87.68 |
| **Natural Language Inference** | | | | | | | | |
| SNLI | **77.23** / **77.69** (0.34) | 0.56 / 0.58 (0.01) | 76.96 / 76.77 (0.07) | 0.12 / 0.12 (0.00) | 76.46 / 76.65 (0.02) | 0.27 / 0.30 (0.00) | 0.27 | 75.35 |
| **Paraphrase Detection** | | | | | | | | |
| QQP | **78.74** / 78.50 (0.46) | 0.59 / 0.57 (0.01) | 78.40 / 78.46 (0.21) | 0.04 / 0.03 (0.00) | 78.61 / **78.54** (0.11) | 0.33 / 0.34 (0.00) | 0.30 | 77.78 |
| **Question Answering** | | | | | | | | |
| bAbI 1 | 99.10 / **100** (0.00) | 0.56 / 0.72 (0.01) | **100.00** / **100** (0.00) | 0.07 / 0.10 (0.00) | 99.90 / 100.00 (0.00) | 0.22 / 0.22 (0.01) | 0.19 | 42.00 |
| bAbI 2 | 40.10 / **49.43** (7.96) | 0.48 / 0.49 (0.12) | 40.20 / 43.50 (5.93) | 0.05 / 0.12 (0.00) | **56.10** / 32.50 (6.13) | 0.21 / 0.17 (0.01) | 0.12 | 33.20 |
| bAbI 3 | 47.70 / 22.90 (0.80) | 0.43 / 0.88 (0.00) | 50.90 / 49.57 (5.44) | 0.10 / 0.10 (0.00) | **51.20** / **50.77** (8.58) | 0.18 / 0.16 (0.01) | 0.07 | 31.60 |

Table 1: Accuracy and conicity of Vanilla, Diversity and Orthogonal LSTMs across different datasets. Authors' results are left, our results including standard deviation between brackets are right. Accuracy of a Multilayered Perceptron (MLP) model and conicity of vectors uniformly distributed with respect to direction is also reported for reference.

## 4.2 Experiment 2: Usefulness of importance ranking of hidden states

The second experiment that was conducted in the paper researched whether attention weights provide a useful importance ranking of hidden representations. To do so, the authors erase the hidden representations in the descending order of the importance (highest attention to lowest) until the model's decision changes. The authors observed that, in several datasets, a large fraction of the representations in the Vanilla LSTM model have to be erased to obtain a decision flip. This observation suggests that representations in the lower end of the importance ranking do play a significant role, which makes the usefulness of attention ranking in Vanilla LSTMs questionable. In contrast, the authors claim that a decision flip is obtained much quicker using their Diversity and Orthogonal LSTM, implying higher importance to higher ranked representations.

In figure 1 we present our results of this experiment. For reference, the authors' results are included in Appendix B. We notice similar trends as those observed by the authors. For the IMDB and 20 News datasets we observe a quicker decision flip for the Diversity and Orthogonal LSTMs when compared to the Vanilla LSTM. For the QQP dataset only the Orthogonal LSTM results in a quicker decision flip, while the authors of the paper find that the Diversity LSTM has a quicker decision flip as well. For the bAbi 1 dataset, all models have an equally fast decision flip, which can be observed in the paper as well. Overall, our results are in line with the authors' results. Therefore it can be stated that, in contrast to a Vanilla LSTM, the top elements of the attention ranking of the Diversity and Orthogonal models are better able to concisely describe the model's decisions. This suggests that their attention weights provide a faithful explanation of the model's performance. Ultimately, these findings support claim 2, which states that the attention distributions of the proposed models offer more transparency as they provide a precise importance ranking of the hidden states.

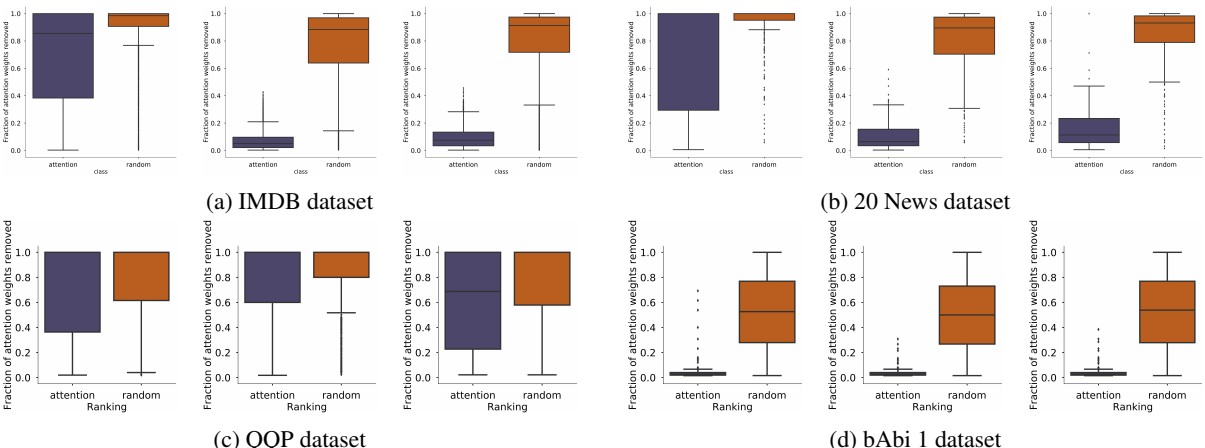

(a) IMDB dataset            (b) 20 News dataset

(c) QQP dataset            (d) bAbi 1 dataset

Figure 1: Box plots of fraction of hidden representations removed for a decision flip for the the Vanilla, Diversity and Orthogonal LSTMs (left to right, respectively).

## 4.3 Experiment 3: Meaningfulness of attention weights

To strengthen the affirmed claim made in experiment 2, which states that the attention weights of the Diversity and Orthogonal LSTM provide more faithful predictions and therefore offer more transparency due to a more precise importance ranking, another experiment was conducted. Here, the authors randomly permuted the attention weights and observed the difference in the model's output. Subsequently, they plotted the median of Total Variation Distance (TVD) between the output distribution before and after the permutation for different values of maximum attention in the Vanilla, Diversity and Orthogonal LSTM models. In the plots stemming from the paper (see appendix B), it can be observed that randomly permuting the attention weights in the Diversity and Orthogonal LSTM model resulted in significantly different outputs. However, there is little change in the Vanilla LSTM model's output for several datasets suggesting that here the attention weights are not so meaningful.

In figure 2 the results of our experiment are presented. While the authors, for unclear reasons, chose to only plot the TVD for one class, we plotted it for both classes. When compared to the authors' results, we see similar trends. For the IMDB dataset, we observe that randomly permuting the attention weights in the Diversity and Orthogonal LSTMs results in significantly different outputs, whereas the output of the Vanilla LSTM does not change much. We observe the same for the 20News dataset. It is noticeable however, that for one class (in blue) of the 20News dataset in combination with the Vanilla LSTM there is almost no difference in outputs (left subfigure in figure 2b), while for the other class (in brown) we observe that there is a considerable amount of permutations in which the output does change. It might be for this reason that the authors chose to only show the TVD for one class of each dataset.

Although our results are not completely in line with the paper and the evidence is not as convincing, it can be said that in general, randomly permuting the attention weights in the Diversity and Orthogonal LSTM models results more often in different outputs than permuting the attention weights in the Vanilla LSTM. The sensitivity to random permutations of the attention weights in the proposed models suggests that they provide a more faithful explanation for the model's predictions. This finding contributes to the previous confirmation of claim 2, which states that Diversity and Orthogonal attention distributions offer more transparency as they provide a more precise importance ranking of the hidden states.

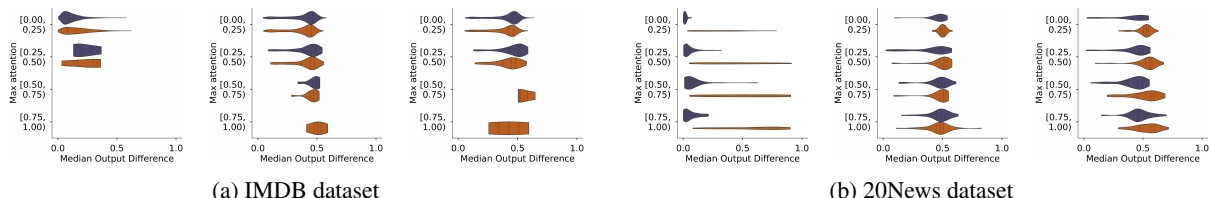

(a) IMDB dataset            (b) 20News dataset

Figure 2: Comparison of Median output difference when randomly permuting the attention weights for the Vanilla, Diversity and Orthogonal LSTM models (left to right, respectively). Plotted for the IMDB and 20News dataset. Blue: Class 0, Brown: Class 1.

### 4.4 Experiment 4: Comparison with Rationales

In the binary classification tasks a single input sequence is used to make a prediction. For those tasks, the authors analyzed how much attention is given to words in the sentence that are important for the prediction. Specifically, a minimum subset of words in the input sentence were selected with which the model can accurately make predictions. This subset of words, called Rationales, were obtained using the REINFORCE algorithm. After obtaining the rationales, the total amount of attention given to this subset is calculated. The authors observed that the accuracy of predictions made from the extracted rationales was within 5% of the accuracy made from the entire sentences, while the LSTM Diversity model provides substantially more attention to rationales than the Vanilla LSTM. This would indicate that the attention weights in the Diversity LSTM are better able to indicate words that are important for making predictions.

In table 2 our averaged results over three runs including standard deviation are presented. For reference, the authors' results have been included in the table as well. While the authors did not incorporate the results of the Orthogonal LSTM, we also added these. We can observe that the results for the Diversity and Orthogonal LSTM are very similar.

In comparison to the authors' results, we notice some differences. For the SST and IMDB datasets it is visible that the Diversity and Orthogonal LSTMs have a relative low rationale length value and and a relative high rationale attention value, which is in line with the results in the paper. For these datasets using the Vanilla LSTM, however, we observe very high values for the rationale length. This indicates that in the Vanilla LSTM almost all words are considered important. Subsequently, it is not unsurprising to see that almost all attention is given to these rationales. This is very different from the results observed in the paper which show a lower rationale attention value and a higher rationale length value in contrast to the Diversity LSTM. However, for the 20News dataset comparable results as in the paper are noticeable.

In most cases, we do not observe that the Diversity LSTM model provides much higher attention to rationales which are shorter than the Vanilla LSTM model's rationales. Thus, we cannot confirm claim 3 which states that the attention weights in the Diversity LSTM are able to better indicate words that are important for making predictions.

| Dataset | Vanilla LSTM | | Diversity LSTM | | Orthogonal LSTM | |
|---|---|---|---|---|---|---|
| | Rationale Attention | Rationale Length | Rationale Attention | Rationale Length | Rationale Attention | Rationale Length |
| SST | 0.348 / 0.89 (0.07) | 0.240 / 0.87 (0.08) | 0.624 / 0.58 (0.08) | 0.175 / 0.21 (0.06) | - / 0.47 (0.04) | - / 0.14 (0.02) |
| IMDB | 0.472 / 0.90 (0.13) | 0.217 / 0.84 (0.21) | 0.761 / 0.87 (0.05) | 0.169 / 0.27 (0.04) | - / 0.68 (0.05) | - / 0.17 (0.01) |
| 20News | 0.627 / 0.66 (0.17) | 0.215 / 0.56 (0.03) | 0.884 / 0.95 (0.02) | 0.173 / 0.27 (0.05) | - / 0.90 (0.04) | - / 0.26 (0.04) |

Table 2: Mean Attention given to the generated rationales with their mean lengths (in fraction). Authors' results are left, our results including standard deviation between brackets are right.

### 4.5 Experiment 5: Comparison with attribution methods

In the fifth experiment, the authors examined how well their attention weights agree with gradient-based attribution methods Sundararajan et al. (2017). For every input word, they computed these attributions and normalized them to obtain a distribution over the input words. Subsequently, they computed the Pearson correlation and JS divergence between the attribution distribution and the attention distribution. The authors observed that attention weights in the Diversity LSTM better agree with gradients with an average (relative) 64.84% increase in Pearson correlation and an average (relative) 17.18% decrease in JS divergence over the Vanilla LSTM across the datasets. If we recalculate these averages due to our usage of less datasets, we obtain an average (relative) 90.81% increase in Pearson correlation and an average (relative) 29.37% decrease in JS divergence over the Vanilla LSTM across datasets. Similar trends follow for the Integrated Gradients attribution method.

Our results regarding the difference of the Diversity and Orthogonal LSTMs with respect to the Vanilla LSTM are shown in table 3. Relative increases or decreases were calculated by comparing the averaged scores over three runs. For reference, these values including standard deviation and the authors' results can be found in Appendix B. We observe an average (relative) increase of 30.96% and 29.80% in Pearson correlation between the Attention weights and the Gradients and Integrated Gradients, respectively, for the Diversity LSTM when compared to the Vanilla LSTM. Although these increases are lower than those found in the paper, the general trend is similar. For the JS divergence, however, we see an average (relative) increase of 8.48% between the Attention weights and Gradients for the Diversity LSTM compared to the Vanilla LSTM. This is not in line with the authors' results which suggest an average (relative) 29.37% decrease in JS divergence. We also do not notice a decrease in JS Divergence between the attention weights and Integrated Gradients as our results show an average (relative) increase of 0.81%.

It is also noticeable that all our averaged increase and decrease results have relatively high standard deviations across the different datasets, which means that results differ significantly per dataset. For example, the Babi 1 and Babi 2 Pearson Correlation between the Attention weights and Gradients for the Diversity LSTM is actually worse than for the regular LSTM (-4.45% and -2.47% respectively) which is not in line with the previously mentioned average (relative) 30.96% increase in Pearson Correlation. These inconsistent results are observable for more datasets. Due to the fluctuations of results between datasets and the average increase of JS divergence when compared to Vanilla LSTMs, our results of this experiment cannot confirm claim 4, which states that the attention distributions of the diversity-driven LSTM models offer more transparency as they correlate better with attribution methods such as gradients and integrated gradients.

| Dataset | Pearson Correlation | | | | JS Divergence | | | |
| | Gradients (Mean incr./decr.) | | Integrated Gradients (Mean incr./decr.) | | Gradients (Mean incr./decr.) | | Integrated Gradients (Mean incr./decr.) | |
| | Diversity | Orthogonal | Diversity | Orthogonal | Diversity | Orthogonal | Diversity | Orthogonal |
|---|---|---|---|---|---|---|---|---|
| **Text Classification** | | | | | | | | |
| SST | +71.33 | +58.67 | +105.00 | +93.33 | -20.00 | +8.57 | -23.08 | -2.56 |
| IMDB | +13.14 | +14.41 | +6.05 | +9.30 | +2.94 | -11.76 | +24.39 | -2.44 |
| 20News | +48.68 | +36.51 | +90.78 | +78.72 | -19.05 | +38.10 | -50.00 | -13.79 |
| **Natural Language Inference** | | | | | | | | |
| SNLI | +5.59 | -7.45 | -3.85 | +5.77 | -11.43 | +5.71 | -4.35 | -8.70 |
| **Paraphrase Detection** | | | | | | | | |
| QQP | +56.70 | +23.71 | +72.92 | +91.67 | +25.93 | +37.04 | +28.13 | +25.00 |
| **Question Answering** | | | | | | | | |
| Babi 1 | -4.45 | +0.68 | +8.55 | +11.11 | +54.17 | +37.50 | -10.29 | -11.76 |
| Babi 2 | -2.47 | -62.96 | -3.70 | -86.42 | +0.83 | +26.45 | -3.27 | +14.38 |
| Babi 3 | +59.20 | -28.80 | -37.35 | -95.18 | +34.48 | +75.86 | 44.92 | +63.65 |
| **Averaged relative difference w.r.t. Vanilla LSTM** | | | | | | | | |
| **Incr/Decr (%)** | +30.96 | +4.35 | +29.80 | +13.54 | +8.48 | +27.18 | +0.81 | +7.96 |
| **Std. deviation** | 28.01 | 28.98 | 44.83 | 55.78 | 22.48 | 19.94 | 23.75 | 19.76 |

Table 3: Relative increase/decrease in % of Pearson correlation and Jensen-Shannon divergence between Attention weights and Gradients/Integrated Gradients for the Diversity and Orthogonal LSTM models w.r.t. Vanilla LSTM.

## 4.6 Additional experiment: Applying LIME framework

In addition to reproducing the experiments from the original paper, we added one extra experiment to verify the claims of the authors. Moreover, this experiment served to further investigate the validity of claim 4, in which it is stated that the attention distributions of the diversity-driven LSTM models correlate better with other attribution methods. To examine and compare the explanations of the different LSTM models, the LIME framework was used. LIME is an abbreviation for local interpretable model-agnostic explanations and was introduced in 2016 as a method to explain how machine learning classifiers and models arrive at their prediction Ribeiro (2016). To do this, the model is treated as a black box, the input instance that needs to be explained is perturbed and a weighted sparse linear model is learned as an explanation around this instance. Therefore, LIME learns a local linear model around the vicinity of this instance. This means that LIME is not able to explain global predictive behaviour of models but can instead be used as an attribution method that describes how models come to their prediction for specific inputs.

In our additional experiment, we took a similar approach as in experiment 5, in which the attention weights distribution was compared with the distribution of gradient-based attribution methods. More specifically, we created LIME explanation objects for instances in the test dataset using the LimeTextExplainer module, which can be used to calculate an importance weight for every word within the test instance. Subsequently, these LIME weight distributions over the test dataset were normalized and compared to the attention weight distributions over the same test instances using the same similarity metrics as in experiment 5. We performed this experiment on four text classification datasets, where we considered all test sentences for the SST and 20News dataset and only the 1000 shortest test sentences for the relatively large and computationally heavy IMDB and Yelp datasets. We averaged results over three runs, the results of which you can find in Appendix B.

We notice similar trends as in experiment 5: on average both the Diversity and Orthogonal LSTM models obtain a higher Pearson correlation compared to the Vanilla LSTM (+37.12% and +31.70%, respectively), while the JS divergence also increases for both models (+20.39% and +24.71%, respectively). Also, high standard deviation across the different datasets, indicating significant differences in results for the different dataset, is similarly to experiment 5 noticeable. Overall, the comparison of the attention weights with the LIME attribution weights produces inconsistent and inconclusive results that cannot substantiate the validity of the claims made by the authors in the original paper.

# 5   Discussion

Within our reproduction study we studied four claims Mohankumar et al. make in their paper titled *towards transparent and explainable attention models* regarding the transparency offered by Diversity and Orthogonal LSTM models in contrast to a regular LSTM model. Our results support claim 1 and 2. No evidence to support claim 3 was found and our evidence to support claim 4 was inconclusive. An additional experiment using LIME was conducted to test claim 4 in an adjusted approach. The results of this experiment were again inconclusive.

Overall, our results across experiments are inconsistent. In experiment 2 and 3 the Diversity and orthogonal LSTM did show the ability of attention weights to provide a more precise importance ranking of hidden states, which in turn offers more faithfulness and transparency. On the other hand, in experiments 4 (comparison with Rationales), 5 (comparison with attribution methods) and our additional LIME experiment no convincing evidence was found showing that the Diversity and Orthogonal LSTM models offer more faithfulness and plausibility.

It is difficult to point out the exact cause of these inconsistencies. The problem could be in the main assumption made by the authors, namely that higher conicity of hidden states (more variety) does not necessarily leads to more transparent models. It could also be a consequence of the design of the experiments. In the Rationale experiment, for example, the REINFORCE algorithm was not able to extract useful Rationales from texts for the Vanilla LSTM as often a large fraction of sentences were labeled as Rationales. Therefore, our results did not support claim 3. In experiment 5 and the LIME experiment we did often notice a higher Pearson correlation for the Diversity and Orthogonal LSTM. At the same time though, we also noticed as larger Jensen-Shannon Divergence for these models. Also, extreme fluctuations in Pearson Correlation and JS Divergence were observed across datasets. It is, due to these mentioned observations, questionable to what extend both similarity metrics are a useful benchmark to compare the Diversity and Orthogonal attention distributions with other attribution methods.

There are two main strengths in our reproduction approach. First of all, in contrast to the authors we also included the results of the Orthogonal LSTM model in the Rationale (4.4) and attribution methods (4.5) experiments. The results in the Rationale experiment showed a very similar performance of both models. Because of the inconclusive outcome of the attribution methods experiment, however, it is difficult to make a good comparison of performance between both models. Also, because these results are left out in the original paper, a comparison between the authors' results and our results is not possible. The second strength of our approach is the fact that we did three runs for each experiment. This made it possible to draw stronger conclusions from findings and detect possible deviations in results as was the case in the attribution Methods experiment. The fact that not all datasets used by the original authors were tested can be considered a weakness in our approach. Considering the fact that all NLP tasks carried out in the original paper were maintained in our study, we do however think that with the subset used, a reflective reproduction study has been carried out. Another shortcoming of our approach is that we did not conduct the qualitative experiments of the original paper, that might have helped to reinforce our conclusions

In conclusion, although we were not able to reproduce everything the authors claim in their paper, we still found some signs that the proposed Diversity and Orthogonal LSTMs could offer something extra in terms of explainability and transparency. Additional research could provide a definitive answer to this question.

## 5.1   What was easy

The authors' code was readily available and included clear instructions how to set up and run all experiments. Also, the authors' arguments to substantiate their claims and the provided results were easy to follow and understand. Verifying a majority of the claims was uncomplicated since all results and plots were automatically created when running the code.

## 5.2   What was difficult

Not all datasets were available and some links to the datasets were outdated. Furthermore, some experiments on particular datasets took quite some time to run, e.g. the Rationale experiment on relative large datasets such as IMDB. Within the code, there were some confusing sections where additional comments could contribute to an easier understanding. In particular, it was confusing that the hidden size of the LSTM models recorded in the configurations file did not match the actual hidden size of the LSTM cells initialized. Lastly, it was unclear how some plots in the paper were created, e.g. the combined display of the median output difference for all LSTM models.

## 5.3   Communication with original authors

There has been no contact with the original authors of the paper.

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

 # 6 A Appendix - Datasets and hyperparameters

| Dataset | Available | Source | Task Specifications |
|---|---|---|---|
| **Binary Classification** | | | |
| Stanford Sentiment Treebank (SST) | Yes | Socher et al. (2013) | Sentiment Analysis with binary target variables (positive/negative) |
| IMDB Movie Reviews | Yes | Maas et al. (2011) | |
| Yelp | Yes | Link | |
| Amazon | No | - | |
| Anemia | No | — | Determining patients type of Anemia (Chronic vs Acute) |
| Diabetes | No | - | Diabetes patient diagnosing |
| 20 Newsgroups (20News) | Yes | Jain and Wallace (2019) | Classify sports-news articles into (baseball/hockey) |
| Tweets | No | Nikfarjam et al. (2015) | Detect if a tweet describes an adverse drug reaction or not |
| **Natural Language Inference** | | | |
| SNLI | Yes | Bowman et al. (2015) | Recognizing textual entailment within sentence pairs |
| **Paraphrase Detection** | | | |
| Quora Question Paraphrase (QQP) | Yes | Wang et al. (2018) | Classify as paraphrased or not |
| **Question Answering** | | | |
| bAbI 1 | Yes | | Answering questions that require 1, 2 or 3 supporting statements from the context |
| bAbI 2 | Yes | Weston et al. (2015) | |
| bAbI 3 | Yes | | |
| CNN News Articles (CNN) | No | - | Question answering |

Table 4: Datsets used in the Mohankumar et al. paper and their current availability, sources and corresponding tasks in this paper.

| Hyperparameter | Vanilla LSTM | Diversity LSTM | Orthogonal LSTM |
|---|---|---|---|
| Word embedding dimension[1] | 300 | 300 | 300 |
| Encoder hidden size[2] | 256 | 256 | 256 |
| Generator hidden size | 256 | 64 | 64 |
| Sparsity lambda | 0.2 | 0.5 | 0.5 |
| Batch size | 32 | 32 | 32 |
| Weight decay | 1e-05 | 1e-05 | 1e-05 |
| Diversity weight | - | 0.5[3] | - |
| Context weight | - | 0 | - |
| Optimizer | Adam | Adam | Adam |
| Learning rate | 0.001 | 0.001 | 0.001 |

Table 5: Table presenting the hyperparameter-setting used in each model

---

[1]Exception for BabI datasets, where word embedding dimension is 50.

[2]Exception for BabI datasets, where encoder hidden size is 128.

[3]Exception for SNLI and CNN datasets, where diversity weights are 0.1 and 0.2 respectively.

 # B Appendix - Experiment results

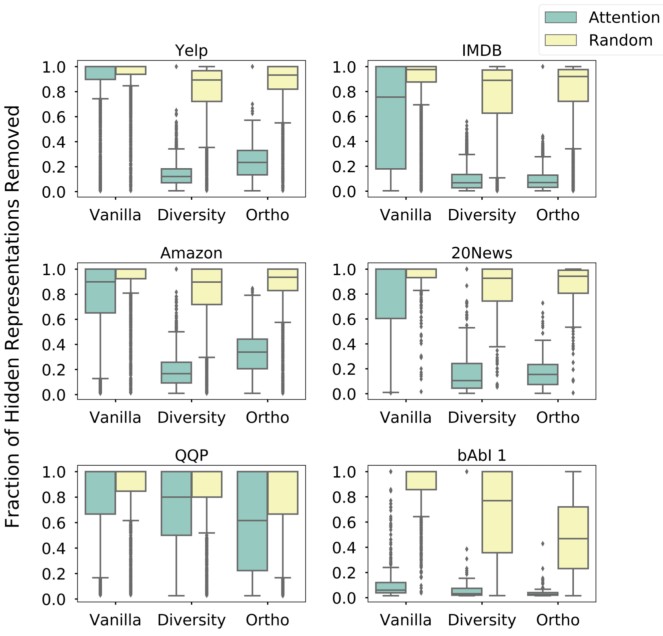

Figure 3: Box plots of fraction of hidden representations removed for a decision flip stemming from the original paper. Dataset and models are mentioned at the top and bottom of figures. Blue and Yellow indicate the attention and random ranking.

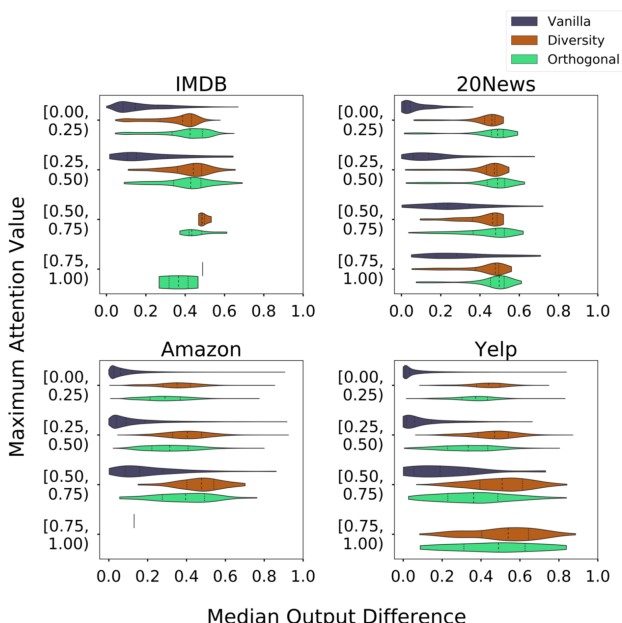

Figure 4: Comparison of Median output difference on randomly permuting the attention weights in the vanilla, Diversity and Orthogonal LSTM models from the original paper. The dataset names are mentioned at the top of each figure. Colors indicate the different models as shown legend

| Dataset | Pearson Correlation ↑ | | | | | | JS Divergence ↓ | | | | | |
|---|---|---|---|---|---|---|---|---|---|---|---|---|
| | Gradients (Mean ± Std.) | | | Integrated Gradients (Mean ± Std.) | | | Gradients (Mean ± Std.) | | | Integrated Gradients (Mean ± Std.) | | |
| | Vanilla | Diversity | Orthogonal | Vanilla | Diversity | Orthogonal | Vanilla | Diversity | Orthogonal | Vanilla | Diversity | Orthogonal |
| **Text Classification** | | | | | | | | | | | | |
| SST | 0.50 (0.08) | 0.86 (0.01) | 0.79 (0.03) | 0.40 (0.07) | 0.82 (0.02) | 0.77 (0.05) | 0.12 (0.02) | 0.09 (0.01) | 0.13 (0.01) | 0.13 (0.02) | 0.10 (0.01) | 0.13 (0.02) |
| IMDB | 0.79 (0.02) | 0.89 (0.01) | 0.90 (0.01) | 0.72 (0.05) | 0.76 (0.03) | 0.78 (0.02) | 0.11 (0.02) | 0.12 (0.01) | 0.10 (0.01) | 0.14 (0.01) | 0.17 (0.02) | 0.13 (0.01) |
| 20News | 0.63 (0.15) | 0.94 (0.01) | 0.86 (0.02) | 0.47 (0.21) | 0.90 (0.01) | 0.84 (0.00) | 0.14 (0.02) | 0.11 (0.01) | 0.19 (0.01) | 0.19 (0.03) | 0.10 (0.00) | 0.17 (0.00) |
| **Natural Language Inference** | | | | | | | | | | | | |
| SNLI | 0.54 (0.02) | 0.57 (0.01) | 0.50 (0.02) | 0.35 (0.02) | 0.33 (0.01) | 0.37 (0.03) | 0.12 (0.00) | 0.10 (0.00) | 0.12 (0.00) | 0.15 (0.00) | 0.15 (0.01) | 0.14 (0.01) |
| **Paraphrase Detection** | | | | | | | | | | | | |
| QQP | 0.32 (0.04) | 0.51 (0.09) | 0.40 (0.05) | 0.16 (0.11) | 0.28 (0.11) | 0.31 (0.02) | 0.09 (0.00) | 0.11 (0.02) | 0.12 (0.00) | 0.11 (0.02) | 0.14 (0.02) | 0.13 (0.01) |
| **Question Answering** | | | | | | | | | | | | |
| Babi 1 | 0.97 (0.01) | 0.93 (0.01) | 0.98 (0.00) | 0.78 (0.01) | 0.85 (0.02) | 0.87 (0.03) | 0.08 (0.01) | 0.12 (0.02) | 0.11 (0.01) | 0.23 (0.02) | 0.20 (0.02) | 0.20 (0.03) |
| Babi 2 | 0.54 (0.01) | 0.53 (0.08) | 0.20 (0.07) | 0.27 (0.11) | 0.26 (0.11) | 0.04 (0.02) | 0.40 (0.01) | 0.41 (0.01) | 0.51 (0.05) | 0.51 (0.05) | 0.49 (0.03) | 0.58 (0.04) |
| Babi 3 | 0.42 (0.02) | 0.66 (0.06) | 0.30 (0.15) | 0.28 (0.07) | 0.17 (0.06) | 0.01 (0.02) | 0.29 (0.02) | 0.39 (0.03) | 0.51 (0.05) | 0.39 (0.05) | 0.57 (0.03) | 0.64 (0.02) |

Table 6: Mean and standard deviation (between brackets) averaged over 3 runs of Pearson correlation and Jensen-Shannon divergence between Attention weights and Gradients/Integrated Gradients for Vanilla, Diversity and Orthogonal LSTM models.

| Dataset | Pearson Correlation ↑ | | | | JS Divergence ↓ | | | |
|---|---|---|---|---|---|---|---|---|
| | Gradients (Mean ± Std.) | | Integrated Gradients (Mean ± Std.) | | Gradients (Mean ± Std.) | | Integrated Gradients (Mean ± Std.) | |
| | Vanilla | Diversity | Vanilla | Diversity | Vanilla | Diversity | Vanilla | Diversity |
| **Text Classification** | | | | | | | | |
| SST | 0.71 ± 0.21 | 0.83 ± 0.19 | 0.62 ± 0.24 | 0.79 ± 0.22 | 0.10 ± 0.04 | 0.08 ± 0.05 | 0.12 ± 0.05 | 0.09 ± 0.05 |
| IMDB | 0.80 ± 0.07 | 0.89 ± 0.04 | 0.68 ± 0.09 | 0.78 ± 0.07 | 0.09 ± 0.02 | 0.09 ± 0.01 | 0.13 ± 0.02 | 0.13 ± 0.02 |
| 20News | 0.72 ± 0.28 | 0.96 ± 0.08 | 0.65 ± 0.32 | 0.67 ± 0.11 | 0.15 ± 0.07 | 0.06 ± 0.04 | 0.21 ± 0.06 | 0.07 ± 0.05 |
| **Natural Language Inference** | | | | | | | | |
| SNLI | 0.58 ± 0.33 | 0.51 ± 0.35 | 0.51 ± 0.38 | 0.26 ± 0.39 | 0.11 ± 0.07 | 0.10 ± 0.06 | 0.16 ± 0.09 | 0.13 ± 0.06 |
| **Paraphrase Detection** | | | | | | | | |
| QQP | 0.19 ± 0.34 | 0.58 ± 0.31 | -0.06 ± 0.34 | 0.21 ± 0.36 | 0.15 ± 0.08 | 0.10 ± 0.05 | 0.19 ± 0.10 | 0.15 ± 0.06 |
| **Question Answering** | | | | | | | | |
| Babi 1 | 0.56 ± 0.34 | 0.91 ± 0.10 | 0.33 ± 0.37 | 0.91 ± 0.16 | 0.33 ± 0.12 | 0.21 ± 0.08 | 0.43 ± 0.13 | 0.24 ± 0.08 |
| Babi 2 | 0.16 ± 0.22 | 0.70 ± 0.13 | 0.05 ± 0.22 | 0.75 ± 0.10 | 0.53 ± 0.09 | 0.23 ± 0.06 | 0.58 ± 0.09 | 0.19 ± 0.05 |
| Babi 3 | 0.39 ± 0.24 | 0.67 ± 0.19 | -0.01 ± 0.08 | 0.47 ± 0.25 | 0.46 ± 0.08 | 0.37 ± 0.07 | 0.64 ± 0.05 | 0.41 ± 0.08 |

Table 7: Mean and standard deviation of Pearson correlation and Jensen-Shannon divergence between Attention weights and Gradients/Integrated Gradients for Vanilla, Diversity and Orthogonal LSTM models from paper.

| Dataset | Pearson Correlation ↑ (Mean + Std.) | | | JS Divergence ↓ (Mean + Std.) | | |
|---|---|---|---|---|---|---|
| | Vanilla | Diversity | Orthogonal | Vanilla | Diversity | Orthogonal |
| SST | 0.41 ± 0.34 | 0.80 ± 0.21 | 0.77 ± 0.24 | 0.13 ± 0.06 | 0.11 ± 0.06 | 0.12 ± 0.06 |
| IMDB | 0.78 ± 0.09 | 0.84 ± 0.09 | 0.82 ± 0.07 | 0.12 ± 0.02 | 0.16 ± 0.04 | 0.16 ± 0.03 |
| 20News | 0.60 ± 0.31 | 0.81 ± 0.16 | 0.73 ± 0.23 | 0.19 ± 0.08 | 0.28 ± 0.11 | 0.29 ± 0.11 |
| Yelp | 0.62 ± 0.29 | 0.68 ± 0.22 | 0.70 ± 0.23 | 0.11 ± 0.07 | 0.13 ± 0.07 | 0.13 ± 0.07 |
| **Averaged relative difference w.r.t. Vanilla LSTM** | | | | | | |
| **Incr/Decr (%)** | - | +37.12 | +31.70 | - | +20.39 | +24.71 |
| **Std. deviation** | - | 35.23 | 32.88 | - | 26.46 | 25.40 |

Table 8: Mean and standard deviation of Pearson correlation and Jensen-Shannon divergence between Attention distributions and the LIME weight distributions for the Vanilla, Diversity and Orthogonal LSTM models implemented on the text classification datasets. The bottom two rows represent the averaged relative difference from the Vanilla LSTM with std. deviation between the datasets.

