# OpenReview forum: "Reproduction study: Towards Transparent and Explainable Attention Models"
_ML_Reproducibility_Challenge/2020 — Reject_

### Official Review · AnonReviewer2 · 2021-02-24
**nonanonymous report**

**Rating:** 1
**Confidence:** 5

**Review:**

The submission is a nonanonymous report, which violates the policy.

As for the detailed reports, it is good and it gives a good reproduction of the original paper.
The authors clearly show what questions they want to answer through this report. According to these lines, the authors conduct experiments and analyses to provide results. With different empirical studies, the authors verify that 1) the prediction performance is almost the same as the original attention mechanism; 2) diversity-driven LSTM indeed gives more transparency attention. However, the other two claims from the original paper are not so convincing or hard to be investigated.
One slight suggestion is the presentation, the last section, Sec 5, the authors could remove the 5.2-4 since they have been shown in the first section.

The score is only raised by the nonanonymous policy, I am sorry for this.

**Familiar With The Original Paper:**

I have not read the original paper

**Reproducibility Summary:**

Report has summary

---

### Official Review · AnonReviewer3 · 2021-03-02
**Good work**

**Rating:** 7
**Confidence:** 4

**Review:**

This paper aims at evaluating the claims made by "Towards Transparent and Explainable Attention Models" by trying to reproduce the experiments in the original paper. Although the authors were not able to run all the experiments in that paper due to some dataset links missing, they have covered a decent amount of datasets and conducted extensive experiments, including an additional LIME experiment to validate their results from a different perspective. This paper is eventually able to validate claims 1 & 2, claim 3 to some extent, but not claim 4: in fact, on some datasets and some metrics (such as JS divergence), this paper observes opposite results.

Pros:
1. This work notices some issues of the presentation of the original paper, namely the fact that the original paper ignored one of the classes when plotting Figure 2 for some reason, which if included seems to cast some shadows over the desirable outcome.  I think this insight is important for future researchers to get a complete understanding of the claims made by the original work.

2. This work runs each experiment multiple times to report the central tendency, which is useful to have.

3. After finding inconsistent results compared to the original work, this paper conducted a further LIME experiment which provides further evidence of the validity of their experiments.

Cons:
1. Given the different observations and the fact that this work used the code from the original work, I think it'd be nice to contact the authors of the original paper.

2. Not being able to evaluate all datasets seems to be a limitation, especially for reporting the mean statistics.

3. In table 3, sometimes both the Pearson correlation and the JS divergence improve. This seems to warrant further investigation: why do they both improve? Is it because the attentions are so spiky that they incur a large penalty when measuring JS?

Questions:
1. It's a bit surprising that in table 2 the vanilla LSTM needs so many words as its rationale, especially for classification tasks with so few classes. Did you check the extracted rationales and see if those are words with strong sentiments? Could it be a bug?

Typos & Presentation Issues:
1. 3.1.3 p -> log p
2. figure 1: caption too small
3. sec 4.4: not unsurprising -> not surprising

Overall, this paper has conducted extensive experiments and repeated most of the experiments in the original paper. This paper noticed some results that seem to be neglected by the original work, and showed some contradicting results from the original work. I think this paper is valuable to researchers who are interested in the original work, and recommend its acceptance.

**Familiar With The Original Paper:**

I have not read the original paper

**Reproducibility Summary:**

Report has summary

---

### Official Review · AnonReviewer1 · 2021-03-11
**Review summary**

**Rating:** 7
**Confidence:** 3

**Review:**

This paper provides a reproducible test of the paper Mohankumar et al. (2020) which claims that current attention mechanisms in LSTM based encoders can neither provide a faithful nor a plausible explanation of the model’s predictions in Natural Language Processing tasks and develop new LSTM units to achieve the goal.

In the reproduction, the authors first give sufficient background information about the paper. Then provide enough experimental details about the dataset used, the training model, and the hyperparameters for optimization. The authors also discuss each experimental result separately and compare the numbers/observations with the original paper.

I feel this piece of investigation is above the acceptance bar and I recommend acceptance.

**Familiar With The Original Paper:**

I have not read the original paper

**Reproducibility Summary:**

Report has summary

---

### Decision · Program_Chairs · 2021-03-31

**Decision:**

Reject

**Comment:**

Overall reviews and/or the paper content not good enough for the AC to recommend to the journal.